# State-Separated SARSA: A Practical Sequential Decision-Making Algorithm with Recovering Rewards

## Abstract

While many multi-armed bandit algorithms assume that rewards for all arms are constant across rounds, this assumption does not hold in many real-world scenarios. This paper considers the setting of recovering bandits (Pike-Burke & Grunewalder, 2019), where the reward depends on the number of rounds elapsed since the last time an arm was pulled. We propose a new reinforcement learning (RL) algorithm tailored to this setting, named the State-Separate SARSA (SS-SARSA) algorithm, which treats rounds as states. The SS-SARSA algorithm achieves efficient learning by reducing the number of state combinations required for Q-learning/SARSA, which often suffers from combinatorial issues for large-scale RL problems. Additionally, it makes minimal assumptions about the reward structure and offers lower computational complexity. Furthermore, we prove asymptotic convergence to an optimal policy under mild assumptions. Simulation studies demonstrate the superior performance of our algorithm across various settings.

## 1 Introduction

The multi-armed bandit (MAB) problem (Lattimore & Szepesvári, 2020) is a sequential decision-making problem between an agent and environment. For each round, the agent pulls an arm from a fixed set and receives a reward from the environment. The objective is to maximize the cumulative rewards over a certain number of rounds. This is equivalent to regret minimization, which is commonly used to evaluate algorithms (Lattimore & Szepesvári, 2020). For superior performance, the key ingredient is an exploration-exploitation tradeoff. During the initial rounds, the agent explores arms at random to gather information about the environment. After that, the agent exploits the knowledge obtained during the exploration phase to choose the best arm.

The MAB framework is widely used and finds applications in various domains (Bouneffouf et al., 2020). For instance, in the context of item recommendation (Gangan et al., 2021), MAB algorithms can be applied by interpreting arms as items and rewards as conversion rates. Another example is dynamic pricing (Misra et al., 2019), where arms and rewards correspond to price and profit, respectively. Moreover, in a language-learning application described in Yancey & Settles (2020), MAB algorithms are employed for push notifications, with arms representing notifications and rewards representing app usage.

Many bandit algorithms assume reward stationarity, meaning constant rewards across rounds (Garivier & Moulines, 2011). In such cases, continuing to draw the arm with the highest expected reward is optimal for cumulative rewards. However, this assumption does not hold in the examples presented above. Instead, rewards often depend on the timing of arm selections. In recommendation systems, for example, commodities should be purchased more frequently than other expensive items. Assuming the reward stationarity, bandit algorithms would repeatedly recommend the same item. On the contrary, the purchase probability would increase if the recommendation is made after suggesting a variety of items. In dynamic pricing, it may be more profitable, in the long run, to discount occasionally than to continue discounting for immediate profit. Similarly, occasional push notifications may be more effective at capturing attention than frequent notifications conveying the same message, supported by (Yancey & Settles, 2020) through offline and online experiments.

To address this situation, we consider the case where the reward function depends on the time elapsed since the last arm was pulled, known as *recovering bandits* (Pike-Burke & Grunewalder, 2019). Various algorithms have been proposed in the MAB framework, but most assume specific reward structures (Yancey & Settles (2020); Simchi-Levi et al. (2021); Kleinberg & Immorlica (2018); Leqi et al. (2021);Moriwaki et al. (2019); Warlop et al. (2018)) or are computationally expensive (Laforgue et al., 2022). Implementing such algorithms for unknown reward functions over a long sequence of rounds would be challenging.

An alternative approach to dealing with the change in rewards is to apply a reinforcement learning (RL) algorithm (Sutton & Barto, 2018), considering *states* as the elapsed rounds for each arm. However, popular tabular RL algorithms, such as Q-learning (Watkins & Dayan, 1992) and SARSA (Rummery & Niranjan, 1994)), face a combinatorial problem in terms of the number of arms and states. Specifically, we need to estimate the Q-function for all possible combinations across the maximum number of states for each arm. Consequently, tabular RL algorithms are computationally prohibitive, except for a small number of arms.

To mitigate the combinatorial issue, this paper proposes a new tabular SARSA, called *State-Separated SARSA* (*SS-SARSA*). We introduce for each arm *State-Separated Q-function* (*SS-Q-function*), which depends on the states for both the associated and a pulled arm, and similarly update them to the standard tabular SARSA. The update thus requires only the states of the two arms. As a result, the number of Q functions to be estimated is significantly reduced, leading to more efficient estimation. Furthermore, this algorithm guarantees convergence to Bellman optimality equation for Q-functions, meaning it achieves an optimal policy asymptotically. Additionally, since our algorithm is a slightly modified version of SARSA, it can be solved in linear time for rounds and is faster than the related work (Laforgue et al., 2022). Also, we introduce a new policy called *Uniform-Explore-First*; during the exploration phase, it pulls the least frequently selected arm for given states to update Q-functions uniformly. Subsequently, the agent pulls arms to maximize cumulative rewards. Note that even random exploration such as $\epsilon$-greedy (Sutton & Barto, 2018) does not update Q-functions uniformly in our setting. Finally, compared to popular RL algorithms and recovering MAB algorithms (Pike-Burke & Grunewalder, 2019), simulation results across various reward settings demonstrate the superiority of our algorithm in terms of cumulative rewards and optimal policy.

The contributions of this work are summarized as follows.

- In the recovering reward setting, the proposed algorithm SS-SARSA can mitigate the combinatorial computation and can be solved in linear time.

- It is theoretically guaranteed that the proposed algorithm obtains optimal policy asymptotically for any reward structure.

- The proposed policy, Uniform-Explore-First, updates each Q-function uniformly for efficient exploration.

- In various settings, simulation results show the superiority of our algorithm in terms of cumulative rewards and optimal policy over related works.

The remainder of this paper is organized as follows. Section 2 reviews related literature and discusses the differences from our work. We define a formal problem setting in Section 3 and the proposed algorithm in Section 4. In Section 5, we present a convergence analysis for the proposed algorithm. Section 6 shows the simulation results and advantages over the related methods. Finally, we state our conclusion in Section 7.

## 2 Related Work

In sequential decision-making problems, two typical approaches are Multi-Armed Bandits (MAB) and Reinforcement Learning (RL). MAB does not use a state in modeling, while RL incorporates a state. Recovering bandit problems have predominantly been addressed in the MAB context, so we mainly survey that area.

In stationary bandits, where expected rewards are constant over time for each arm, algorithms aim to choose the arm with the highest expected value to maximize cumulative rewards. Numerous algorithms have been proposed for both parametric and nonparametric reward settings, such as KL-UCB (Lai et al., 1985; Garivier

& Cappé, 2011), Thompson sampling (Thompson, 1933; Chapelle & Li, 2011), $\epsilon$-greedy (Sutton & Barto, 2018), and UCB1 (Auer et al., 2002). However, this setting ignores reward changes, a crucial aspect addressed in this paper.

The non-stationary bandit problem considers scenarios where rewards can change over time. Restless bandits allow rewards to vary over rounds independently of the history of pulled arms, incorporating settings like piecewise stationary rewards (Garivier & Moulines, 2011; Liu et al., 2018) and variation budgets (Besbes et al., 2014; Russac et al., 2019). However, even in these settings, the variation of rewards due to the history of pulled arms is not accounted for.

In contrast, Rested Bandits involve rewards that depend on the history of the arms pulled. One of the typical settings is that each arm's reward changes monotonically each time the arm is pulled (Heidari et al., 2016). Examples include Rotting bandits (Levine et al., 2017; Seznec et al., 2019), handling monotonically decreasing rewards, and Rising bandits (Li et al., 2020; Metelli et al., 2022), dealing with monotonic increase.

Another setting in Rested Bandits is *Recovering bandits* (Pike-Burke & Grunewalder, 2019) (or *Recharging bandits* (Kleinberg & Immorlica, 2018)), where rewards depend on the elapsed rounds since the last pull for each arm. Numerous algorithms have been proposed in this context, with many assuming a monotonous increase in rewards as rounds progress (Yancey & Settles (2020); Simchi-Levi et al. (2021); Kleinberg & Immorlica (2018)). An alternative approach involves functional approximation with past actions as contexts, exemplified by stochastic bandits with time-invariant linear dynamical systems (Leqi et al., 2021), contextual bandits (Moriwaki et al., 2019), and linear-approximation RL (Warlop et al., 2018). In contrast to these approaches, we propose an algorithm that does not rely on a specific structure of the rewards.

Several articles make fewer assumptions about rewards (Laforgue et al., 2022; Pike-Burke & Grunewalder, 2019). Laforgue et al. (2022) propose an algorithm based on Combinatorial Semi-Bandits, applying integer linear programming for each block of prespecified rounds to determine the sequence of pulling arms. However, its time complexity is more than quadratic in total rounds, making it impractical for large total rounds. In contrast, our algorithm, a modified version of SARSA, can be computed in linear time.

Pike-Burke & Grunewalder (2019) uses Gaussian process (GP) regression and proves the Bayesian sublinear regret bound without reward monotonicity and concavity. However, their algorithms only considered short-term lookahead and did not guarantee to achieve the optimal policy in the long run, as deemed in RL. In contrast, our RL approach can realize the optimal policy asymptotically.

## 3 Problem Setting

In this section, we introduce recovering bandits (Pike-Burke & Grunewalder, 2019) within the framework of the Markov Decision Process (MDP) to facilitate RL algorithms. The (discounted) MDP is defined as $\mathcal{M} = (\mathcal{A}, \mathcal{S}, f, r, \gamma)$, where $\mathcal{A} = [K] := \{1, 2, \cdots, K\}$ is the index set of $K$ arms[1], $\mathcal{S}^k := [s_{\max}]$ denotes the states of the arm $k$ $(k = 1, \ldots, K)$, and $\mathcal{S} = \prod_{k=1}^{K} \mathcal{S}^k$ is their direct product. Additionally, we let $f_k : \mathcal{S}^k \times \mathcal{A} \to \mathcal{S}^k$ denote a deterministic state transition function for arm $k$, and $f = (f_1, f_2, \cdots, f_K)$ a bundled vector. The stochastic bounded reward function is denoted by $r : \mathcal{S} \times \mathcal{A} \to \mathbb{R}$, and $\gamma \in (0, 1]$ serves as a discount rate.

Key distinctions from standard MDP lie in the state structure and the state transition. In the $K$-dimensional state $\mathbf{s} := (s_1, s_2, \cdots, s_k, \cdots, s_K)$, the component $s_k \in [s_{\max}]$ signifies the elapsed number of rounds for the arm $k$ since its last pull. We limit $s_k$ at $s_{\max}$ even if it surpasses $s_{\max}$ rounds. Consequently, the cardinality of the states in $\mathbf{s}$ is $s_{\max}^K$. With the multi-dimensional states, an agent interacts with an environment over $T$ (allowing $\infty$) rounds as follows. At round $t \in [T]$, given state $\mathbf{s}_t := (s_{t,1}, s_{t,2}, \cdots, s_{t,k}, \cdots, s_{t,K})$, the agent draws an arm $a_t$ from the $K$ arms. This choice is governed by a policy $\pi_t(a_t|\mathbf{s}_t)$, which is a map from $\mathcal{S}$ to $\Delta\mathcal{A}$, the probability distributions on the $K$ arms. The environment then returns a stochastic reward $r(s_{t,a_t}, a_t)$, which depends on only $a_t$ and the corresponding state $s_{t,a_t}$. Note that $r$ is independent of the

---

[1]In the context of RL, arms are often referred to as actions, but we use the term "arm" following the original recovering bandits (Pike-Burke & Grunewalder, 2019).

other arm states. Finally, the next state $\mathbf{s}_{t+1}$ is updated as $s_{t+1,k} = f(s_{t,k}, a_t) := \min\{s_{t,k} + 1, s_{\max}\}$ for $k \neq a_t$ and $:= 1$ for $k = a_t$, as stated in the previous paragraph.

With the above MDP, our goal is to maximize the expected (discounted) cumulative rewards, which is defined by

$$V_\pi(\mathbf{s}) := \mathbb{E}_\pi\left[\sum_{t=0}^{T} \gamma^t r(s_{t,a_t}, a_t) \mid \mathbf{s}_0 = \mathbf{s}\right],$$

where $\pi$ is a given stationary policy, which does not depend on time $t$ and $\mathbf{s}$ is an initial state. The optimal policy is defined as $\pi$ that maximizes $V_\pi(\mathbf{s})$ for any initial state. In our MDP, when $T = \infty$ and $\gamma < 1$ (i.e. infinite-horizon discounted MDP), it is known (Puterman, 2014) that there exists an optimal policy $\pi^*$, which is stationary and deterministic, meaning that $\pi^*$ is invariant over time, and for any $\mathbf{s} \in \mathcal{S}$, $\pi(a|\mathbf{s}) = 1$ for some $a \in \mathcal{A}$.

## 4  Algorithm

This section presents a novel algorithm to learn the optimal policy. Section 4.1 introduces Q-functions called the State-Separated Q-functions (SS-Q-functions), considering the state structure. Using these SS-Q-functions, Section 4.2 proposes the State-Separated SARSA (SS-SARSA) algorithm for efficient learning. This section focuses on the discounted MDP with infinite horizons (i.e. $T = \infty$ and $\gamma \in (0, 1)$).

### 4.1  State-Separated Q-function: Constructing the MDP with reduced state combinations

We start with the problem of the tabular RL approach: the combinatorial explosion associated with the Q-function

$$Q(\mathbf{s}, a) := \mathbb{E}_\pi\left[\sum_{t=0}^{T} \gamma^t r(s_t^{a_t}, a_t) \mid \mathbf{s}_0 = \mathbf{s}, a_0 = a\right]. \tag{1}$$

(1) can also be expressed in Bellman-equation form (Sutton & Barto, 2018):

$$Q(\mathbf{s}, a) = \mathbb{E}_r[r(s_a, a)] + \gamma Q(\mathbf{s}', a'). \tag{2}$$

Here, $\mathbf{s}'$ and $a'$ represent the next state and arm after $\mathbf{s}$ and $a$, respectively; $\mathbf{s}' = f(\mathbf{s}, a)$ and $a' \sim \pi(\cdot|\mathbf{s}')$. We also define the Bellman optimality equation for Q-function as

$$Q^*(\mathbf{s}, a) = \mathbb{E}_r[r(s_a, a)] + \gamma \max_{a' \in \mathcal{A}} Q^*(\mathbf{s}', a'), \tag{3}$$

where $Q^*$ is the Q-function concerning the optimal policy.

Unlike the MDP with a probabilistic state transition, there is no need to take the expectation of $\mathbf{s}'$ in the second term of (3). Then, since the cardinality of $\mathbf{s}$ is $s_{\max}^K$, tabular Q-learning (Watkins & Dayan, 1992)/ SARSA (Rummery & Niranjan, 1994) has to estimate the Q-function for $s_{\max}^K \times K$ combinations of the argument. These algorithms are updated with the following rules respectively:

$$\text{Q-learning:} \quad \hat{Q}(\mathbf{s}, a) \leftarrow \hat{Q}(\mathbf{s}, a) + \alpha\left(r(s_a, a) + \gamma \max_{a'} \hat{Q}(\mathbf{s}', a') - \hat{Q}(\mathbf{s}, a)\right) \tag{4}$$

$$\text{SARSA:} \quad \hat{Q}(\mathbf{s}, a) \leftarrow \hat{Q}(\mathbf{s}, a) + \alpha\left(r(s_a, a) + \gamma \hat{Q}(\mathbf{s}', a') - \hat{Q}(\mathbf{s}, a)\right) \tag{5}$$

Thus, these algorithms must learn for a combinatorial number of states, which is prohibitive unless $s_{\max}$ and $K$ are small.

To mitigate this computational difficulty, we introduce SS-Q-functions. Combining these new Q-functions results in a significant reduction in state combinations compared to the original Q-functions.

More specifically, *State-Separated Q-function (SS-Q-function)* is defined by a form of Bellman-equation

$$Q_{SS,k}(s_k, s_a, a) := \mathbb{E}_r[r(s_a, a)] + \gamma Q_{SS,k}(s'_k, s'_{a'}, a') \tag{6}$$

for each $k \in [K]$ and $a \in [K]$. It is similar to the Bellman equation of the original Q-function but involves only two-dimensional states; it depends solely on the state $s_k$ and the state of the pulled arm $s_a$.

We can recover the original Q-function by aggregating these new Q-functions. For a fixed $a \in [K]$, by adding (6) over all $k \in [K]$ and dividing it by $K$,

$$\frac{1}{K}\sum_{k=1}^{K} Q_{SS,k}(s_k, s_a, a) = \mathbb{E}_r[r(s_a, a)] + \gamma\frac{1}{K}\sum_{k=1}^{K} Q_{SS,k}(s'_k, s'_{a'}, a'). \tag{7}$$

Since $r(s_a, a)$ is independent of $k \in [K]$, the instantaneous reward remains unchanged even after the aggregation. Let the left-hand side of (7) be denoted by $Q(\mathbf{s}, a)$, i.e.

$$Q(\mathbf{s}, a) := \frac{1}{K}\sum_{k=1}^{K} Q_{SS,k}(s_k, s_a, a)$$

for each $\mathbf{s} \in [s_{max}]^K$ and $a \in [K]$. Then (7) is equivalent to

$$Q(\mathbf{s}, a) = \mathbb{E}_r[r(s_a, a)] + \gamma Q(\mathbf{s}', a'), \tag{8}$$

which coincides with the definition of the original Q-function. Note that computation with SS-Q-functions requires only $s_{max}^2 K^2$ variables, while the naive implementation of the original Q-function needs $s_{\max}^K \times K$. Similarly, we can also construct the Bellman-optimal equation by aggregating SS-Q-functions.

## 4.2 State-Separated SARSA: A novel efficient RL algorithm in recovering bandits

We introduce *State-Separated SARSA (SS-SARSA)* building on the SS-Q-functions and explain its advantages in comparison to conventional tabular RL and MAB approaches. We also propose a policy tailored to our MDP, which achieves efficient uniform exploration across all the variables of the Q-function.

To begin, we outline SS-SARSA, illustrated in Algorithm 1. Given input parameters $T$, $\gamma$, and initial states $\mathbf{s}_0$, the estimates of SS-Q-functions $\hat{Q}_{SS,k}(s_k, s_a, a)$ are initialized to zero. The algorithm then proceeds in the following way. The agent pulls an arm $a$ from $K$ arms according to a proposed policy, *Uniform-Explore-First $\pi$*, which will be discussed later. Following the state transition rule described in Chapter 3, the next states $\mathbf{s}'$ transition to one where $k = a$, and for the states of other arms, they transition to $\min\{s+1, s_{\max}\}$. Then, for the pulled arm $a$ and for all $k \in [K]$ the SS-Q-functions are updated as follows:

$$\hat{Q}_{SS,k}(s_k, s_a, a) \leftarrow \hat{Q}_{SS,k}(s_k, s_a, a) + \alpha(r(s_a, a) + \gamma\hat{Q}_{SS,k}(s'_k, s'_{a'}, a') - \hat{Q}_{SS,k}(s_k, s_a, a)), \tag{9}$$

where $\alpha \in [0, 1]$ represents a learning rate that is independent of $(s_k, s_a, a)$. The above expression is analogous to the update rule in tabular SARSA given in (5). By defining $\hat{Q}(\mathbf{s}, a) := \frac{1}{K}\sum_{k=1}^{K}\hat{Q}_{SS,k}(s_k, s_a, a)$ and combining as in (7) and (8) using these estimated SS-Q-functions, we can update

$$\hat{Q}(\mathbf{s}, a) \leftarrow \hat{Q}(\mathbf{s}, a) + \alpha\frac{1}{K}\sum_{k=1}^{K}(r(s_a, a) + \gamma\hat{Q}_{SS,k}(s'_k, s'_{a'}, a') - \hat{Q}_{SS,k}(s_k, s_a, a)). \tag{10}$$

$$\iff \hat{Q}(\mathbf{s}, a) \leftarrow \hat{Q}(\mathbf{s}, a) + \alpha\left(r(s_a, a) + \gamma\hat{Q}(\mathbf{s}', a') - \hat{Q}(\mathbf{s}, a)\right) \tag{11}$$

,which has the same form as SARSA (5).

Compared to the related works, SS-SARSA has three advantages: reduced combinations of the estimated SS-Q-functions, low time complexity, and long-term lookahead. First, the combination of SS-Q functions to be estimated is at most $s_{max}^2 K^2$, which is significantly smaller than $s_{max}^K$ of Q-functions in tabular RL

algorithms. Second, our algorithm updates $K$ SS-Q-functions per round, similar to tabular SARSA, thus its time complexity is just $\mathcal{O}(KT)$. It has better computational efficiency compared to the $\mathcal{O}(K^{5/2}T^{9/4})$ time complexity associated with regret guarantees in Laforgue et al. (2022). Finally, our RL approach aims to identify the optimal policy over the entire duration of total rounds. In contrast, MAB approach determines the optimal sequence of selected arms only for short rounds (Laforgue et al., 2022; Pike-Burke & Grunewalder, 2019).

**Remark 4.1.** *(Why does our algorithm adopt SARSA update rule instead of Q-learning update?) Our algorithm updates the SS-Q-functions using the SARSA update rule (5) and combine these SS-Q-functions (10). Another possibility would be to use Q-learning (4). However, it would not fit our setting due to the max operator. In fact, if we apply (4) to each $Q_{SS,k}$ and combine them with the learning rate $\alpha$, the max operator part becomes*

$$\frac{1}{K}\sum_{k=1}^{K}\alpha \max_{a'} Q_{SS,k}(s_k', s_{a'}', a'), \tag{12}$$

*which should be $\alpha \max_{a'} \frac{1}{K}\sum_{k=1}^{K} Q_{SS,k}(s_k', s_{a'}', a') = \alpha \max_{a'} Q(\mathbf{s}', a')$ to update the form of Q-learning. Therefore convergence to the optimal policy would be questionable.*

Before introducing our new policy, we discuss why random exploration doesn't work well in our MDP. Under random exploration, we can compute the probability that arm $k$ is pulled at state $s_k = i$ in round $t$, denoted by $p_{t,i,k}$. Since $\pi(a|\mathbf{s}) = \frac{1}{K}$ under a random policy, we have $p_{t,i,k} = \frac{1}{K} \times q_{t,i,k}$, where $q_{t,i,k}$ is the probability that arm $k$ is in state $i$ at round $t$. When $t \geq s_{\max}$, the probability $p_{t,i,k}$ does not depend on $t$ [2]. The probability for each state is as follows. Since for any $s_k \in [s_{\max}]$ the state of arm $k$ transitions to one when pulling arm $k$, by the total law of probability, $q_{t,1,k} = \frac{1}{K}\sum_{j=1}^{s_{\max}} q_{t-1,j,k} = \frac{1}{K}$ and thus $p_{t,1,k} = \frac{1}{K^2}$. For $i = 2, 3, \cdots, s_{\max} - 1$, an arm reaches state $i$ exactly when arm $k$ is not pulled at state $(i-1)$, which means $q_{t,i,k} = (1 - \frac{1}{K})q_{t-1,i-1,k}$. Thus, $q_{t,i,k} = (1 - \frac{1}{K})^{i-1}\frac{1}{K}$, and $p_{t,i,k} = (1 - \frac{1}{K})^{i-1}\frac{1}{K^2}$. For $i = s_{\max}$, by the law of total probability, $p_{t,s_{\max},k} = 1 - \sum_{j=1}^{s_{\max}-1} p_{t,j,k} = 1 - \frac{1}{K^2} - \sum_{i=2}^{s_{\max}-1}(1 - \frac{1}{K})^{i-1}\frac{1}{K^2}$, which is strictly more than $\frac{1}{K^2}$ because $p_{t,1,k} = \frac{1}{K^2}$ and $p_{t,i,k} < \frac{1}{K^2}$ for $i = 2, 3, \cdots s_{\max} - 1$.

This result implies that even when applying policies that randomly select an arm given a state (e.g., random exploration with a positive probability $\epsilon$ in $\epsilon$-greedy (Sutton & Barto, 2018)), SS-Q-functions are frequently updated at $s_{\max}$ compared to other states. This property is notable in the case where $s_{\max} < K$. As a numerical example, when $K = s_{\max} = 6$, for any $t$ and $k$, $p_{t,s_{\max},k}, \approx 0.067$ and $p_{t,s_{\max-1},k} \approx 0.013$. In contrast, when $K = 6$ and $s_{\max} = 3$, for any $t$ and $k$, $p_{t,s_{\max},k} \approx 0.116$ and $p_{t,s_{\max-1},k} \approx 0.023$. Variation in the frequency of these updates has a negative impact, especially when the state with the highest reward is not $s_{\max}$.

In this paper, we propose an alternative policy, named *Uniform-Explore-First*, which follows a strategy of exploring information uniformly initially and subsequently exploiting this explored information. Specifically, during the specified initial rounds $E$, the agent explores the arm with the minimum number of visits over the $(\mathbf{s}, a)$ pairs, given the current state $\mathbf{s}$ up to round $t$, denoted as $v_t(\mathbf{s}, a)$. If $v_t(\mathbf{s}, a)$ is tied for multiple arms, the selection of the arm can be arbitrary. After the exploration phase, the agent adopts a greedy approach by pulling the arm with the maximum $\hat{Q}(\mathbf{s}, a)$ to maximize the (discounted) cumulative rewards. In contrast to random exploration, our exploration strategy uniformly pulls arms at states other than $s_{\max}$.

## 5 Convergence Analysis

This section presents a convergence analysis and provides remarks on our problem setting. Throughout this section, our focus is on the infinite-horizon discounted MDP, i.e., $T = \infty$ and $\gamma < 1$.

**Theorem 5.1.** *(Convergence of Q-functions) Suppose that the variance of the (stochastic) reward $r$ is finite and $\alpha_t = \frac{1}{t+t_0}$ where $t_0$ is some constant value. This learning rate satisfies Robbins-Monro scheme*

---

[2]If $t < s_{\max}$, some states cannot be reached. For example, if $s_{0,k} = s_{\max}$, it takes at least $s_{\max}$ rounds to visit the state at $s_k = s_{\max} - 1$.

---

**Algorithm 1** State-Separated SARSA (SS-SARSA)

---

**Input:** $T$, $\gamma$, $\alpha$
**Initialize:** $\hat{Q}_{SS,k}(s_k, s_a, a) \leftarrow 0$ for all $k \in [K]$ and $a \in [K]$
**for** $t = 1, 2, \cdots, T$ **do**

$$a \leftarrow \begin{cases} argmin_{a \in [K]} v_t(\mathbf{s}, a) \text{ when } t \leq E \\ argmax_{a \in [K]} \hat{Q}_t(\mathbf{s}, a) \text{ when } t > E \end{cases} \qquad \triangleright \text{ Uniform-Explore-First}$$

$r \leftarrow r(s_a, a)$

$$s'_k \leftarrow \begin{cases} 1 & \text{for } k = a \\ \min\{s_k + 1, s_{\max}\} & \text{for } k \in [K] \backslash \{a\} \end{cases}$$

Update $\hat{Q}_{SS,k}(s_k, s_a, a)$ for all $k \in [K]$:

$\hat{Q}_{SS,k}(s_k, s_a, a) \leftarrow \hat{Q}_{SS,k}(s_k, s_a, a) + \alpha(r(s_a, a) + \gamma \hat{Q}_{SS,k}(s'_k, s'_{a'}, a') - \hat{Q}_{SS,k}(s_k, s_a, a))$
**end for**

---

(i.e. $\sum_{t=1}^{\infty} \alpha_t = \infty$ and $\sum_{t=1}^{\infty} \alpha_t^2 < \infty$). Suppose that $\pi_t(\mathbf{s}|a)$ is a GLIE policy, that is, it visits each $(\mathbf{s}, a)$ infinitely often and chooses an arm greedily with probability one in the limit (i.e. $\pi_t(\mathbf{s}|a) = \arg\max_a \hat{Q}(\mathbf{s}, a)$ w.p. 1 as $t \to \infty$). Also, for each $(\mathbf{s}, a)$ pair, define the error of the Q-function at round $t \in [T]$ as $Q_{err,t}(\mathbf{s}, a) := |\hat{Q}_t(\mathbf{s}, a) - Q^*(\mathbf{s}, a)|$. Define a set $V := \{(\mathbf{s}, a)| \text{ visit } (\mathbf{s}, a) \text{ for some policy }\}$. Then, for any $(\mathbf{s}, a) \in V$, $Q_{err,t}(\mathbf{s}, a) \to 0$ with probability 1 as $T \to \infty$.

*Proof:* As indicated in Section 4, since we adopt the learning rate which is independent of $(s_k, s_a, a)$, SS-SARSA can be considered SARSA after combining the SS-Q functions. Consequently, we can prove the convergence theorem analogous to Singh et al. (2000), under the same assumption as SARSA.

**Remark 5.2.** *(No visit at some states) While the convergence theorem in MDP (Watkins & Dayan, 1992; Singh et al., 2000) assumes infinite visits for each $(\mathbf{s}, a)$ pair, in our MDP settings certain states are never visited for any policy. For instance, $(\mathbf{s}, a)$ with $s_k = 1$ for multiple $k$ never appears, since it means multiple arms were pulled at the last time point. Only the value of Q-function within $V$ is needed to learn a policy.*

**Remark 5.3.** *(Convergence theorem for Uniform-Explore-First) The Uniform-Explore-First policy is considered GLIE because it greedily pulls an arm after the exploration phase. However, due to the separation of the exploration and the exploitation phase, our policy does not guarantee the infinite updating of all Q-functions in $V$. Nevertheless, in practice, by taking sufficient rounds for uniform exploration, the estimated Q-function approaches the Bellman optimality equation for Q-function of each $(\mathbf{s}, a) \in V$.*

## 6 Experiments

In this section, we present simulation results to empirically verify the performance of our algorithm. Section 6.1 gives the simulation settings, and Section 6.2 shows the superiority of our algorithm in some metrics.

### 6.1 Simulation Settings

In this subsection, we state MDP environments, metrics, SS-SARSA parameters, and competing algorithms related to our work.

**Initial state:** Throughout the simulation, initial state is assumed to be $\mathbf{s}_{\max}$ for every $k \in [K]$. In item recommendation, if the conversion rate increases with the number of states, representing the freshness of items, this assumption reflects the agent before the item recommendation is exposed. Note that this assumption is made purely for simplicity and that our algorithm can be generalized to any other initial states.

**Discount rate:** We conduct simulations in both discounted ($\gamma \in (0, 1)$) and non-discounted ($\gamma = 1$) settings. The discounted setting is commonly employed in infinite horizon MDP. We set $\gamma$ to $(1 - 10^{-T})$, depending on T, because it makes the rewards in later rounds not significantly smaller. On the other hand, the non-discounted setting is the standard setting in MAB. As discussed in the previous section, our algorithm does

not satisfy the assumption for the convergence theorem. Nevertheless, we will verify that our algorithm performs well in practice for both situations.

**Reward:** To begin, we demonstrate with a small state combination, specifically setting $K = s_{\max} = 3$ and $T = 10^5$. The first arm is nonstationary, with an expected reward of 0.1 for $s_1 \in \{1, 2\}$ and an increase to 0.3 at $s_1 = 3$. Conversely, the remaining two arms are stationary, with an expected reward of 0.2 irrespective of the state of each arm. Note that our claim of superiority is against MAB algorithms rather than the tabular RL algorithms. Indeed, the cardinality of SS-Q functions in SS-SARSA and the Q-function for Q-learning/SARSA are the same.

Next, we address larger-scale problems and provide a comprehensive framework for all remaining experiments. We consider various settings involved in reward heterogeneity and reward increments.

For reward heterogeneity, we consider $K_{\text{best}}$ best arms and the remaining $(K - K_{\text{best}})$ sub-best arms. When $K = K_{\text{best}}$, arms are homogeneous: changes in expected rewards per state increment for all arms are the same, as illustrated in 6-Homo and 10-Homo in Table 1. In contrast, when $K > K_{\text{best}}$, arms are heterogeneous: changes in expected rewards per state increment differ for best arms and sub-best arms, as illustrated in 6-Hetero and 10-Hetero in Table 1.

For reward increments, we consider two cases of changes in expected rewards with state increments: monotone-increasing rewards and increasing-then-decreasing rewards. In the monotone-increasing case, the expected reward for each arm is as follows: for the best arms, the expected rewards start at 0.1 for $s_a = 1$ and increase by $V_{\text{best}}$ per state increment, reaching 0.6 at $s_{\max}$. For the sub-best arms, the expected rewards at $s_a = 1$ remain the same, but increase only by $V_{\text{sub-best}}(< V_{\text{best}})$ per state increment, reaching 0.5 at $s_{\max}$.

The increasing-then-decreasing case is similar to the monotone case, but the state of peak reward differs. Specifically, for the best arms, the expected rewards at $s_a = 1$ are the same as in the monotone case and increase by $V_{\text{best}}$ per state increment, but reach 0.6 at $s_{\max} - 1$ and decrease by $V_{\text{best}}$ at $s_{\max}$. Similarly, for the sub-best arms, the reward at the peak state is 0.5 at $s_{\max} - 1$ and decreases by $V_{\text{sub-best}}$ at $s_{\max}$.

Finally, we state the total rounds and reward distributions. Considering the cardinality of the states, $T = 10^5$ for $K = 3, 6$ and $T = 10^6$ for $K = 10$. Moreover, we use Bernoulli and normal distributions as reward distributions for all simulations. In the normal case, the variance is 0.5. The normal case is more difficult due to its higher variance compared to the Bernoulli case.

| Name | $K$ | $s_{\max}$ | $|\mathbf{s}|$ | $T$ | $K_{\text{best}}$ | $V_{\text{best}}$ | $V_{\text{sub-best}}$ |
|---|---|---|---|---|---|---|---|
| 6-Hetero | 6 | 3 | $3^6 (> 7 \times 10^2)$ | $10^5$ | 3 | $\frac{1}{4}\left(\frac{1}{2}\right)$ | $\frac{1}{5}\left(\frac{2}{5}\right)$ |
| 6-Homo | 6 | 6 | $6^6 (> 4.6 \times 10^4)$ | $10^5$ | 6 | $\frac{1}{10}\left(\frac{1}{8}\right)$ | $-$ |
| 10-Hetero | 10 | 5 | $5^{10} (> 9 \times 10^6)$ | $10^6$ | 5 | $\frac{1}{8}\left(\frac{1}{6}\right)$ | $\frac{1}{10}\left(\frac{2}{15}\right)$ |
| 10-Homo | 10 | 10 | $10^{10}$ | $10^6$ | 10 | $\frac{1}{18}\left(\frac{1}{16}\right)$ | $-$ |

Table 1: Monotone-increasing (increasing-then-decreasing) reward settings (The values enclosed in parentheses in $V_{\text{best}}$ and $V_{\text{sub-best}}$ pertain to the increasing-then-decreasing case.)

**Metrics:** We introduce some metrics for comparing algorithm performance. The first metric is cumulative regret, which is the difference in (discounted) cumulative expected rewards between the best and learning policies. This metric measures the closeness of the learning policy to the optimal one, so smaller is better, and is often used in MAB (Lattimore & Szepesvári, 2020).

If we can define the optimal policy, we also use the rate of optimal policy, which measures how often the optimal policy is obtained out of a thousand simulations. The optimal policy is defined as having no regret during the last $3 \times s_{\max}$ rounds, making it easy to assess the quality of exploration. Note that this metric only reflects the performance after exploration, while cumulative regret/rewards cover both the exploration and exploitation phases. Therefore, even if some algorithms have smaller regret than others, they may not necessarily achieve optimal policy.

In the (first) small-scale and the monotone-increasing case, the optimal policy can be obtained explicitly. In the former case, the optimal policy is to select the first arm only when $s_1 = s_{\max} = 3$ and to choose the

other arms otherwise. In the monotone-increasing case, as long as $s_{\max} \leq K_{\text{best}}$, the agent has the chance to pull the best arm at $s_{\max}$ given the initial state $\mathbf{s}_0 = \mathbf{s}_{\max}$. Therefore, the optimal policy cyclically pulls only $K_{\text{best}}$ best arms at $s_{\max}$.

However, in the increasing-then-decreasing rewards, given $\mathbf{s}_0 = \mathbf{s}_{\max}$, there is no chance to pull the best arm at $s_{\max} - 1$ in the first round. Thus, optimal policy is not trivial.[3] In such a case, we use (discounted) cumulative expected rewards without optimal policy as another metric. The larger the metric, the better, although its maximization is equivalent to the minimization of cumulative regret.

**SS-SARSA parameters:** We set the learning rate and the exploration horizon in our algorithm. As pointed out in the previous section, we use $\alpha_t = \frac{1}{t+t_0}$. The smaller the $t_0$ value, the greater the effect of the update in the early visit, causing unstable learning due to the stochasticity of rewards. Thus, a large $t_0$ is better for mitigating this issue, and we set $t_0 = 5000$ through the simulations.

The size of the exploration should be determined by considering the trade-off between exploration and exploitation. Over-exploration, characterized by uniformly pulling arms in a large fraction of rounds, fails to sufficiently exploit the best arm, leading to large regret or small cumulative rewards. Conversely, under-exploration, characterized by uniformly pulling arms in a small fraction of rounds, fails to gather adequate information about each Q-function, resulting in a low probability of selecting the best arm. Taking this tradeoff into account, we allocate uniform exploration to 10% of the total rounds. (i.e. $E = 0.1T$).

**Compared algorithms:** We introduce other algorithms to compare their performance to our algorithm. The first algorithms are the original tabular model-free RL algorithms: Q-learning (Watkins & Dayan, 1992) and SARSA (Rummery & Niranjan, 1994). These algorithms also have convergence results to the Bellman optimality equation for Q-function. However, in our setting, as described in section 4.1, these algorithms have to estimate enormous Q-functions unless $s_{\max}$ and $K$ are small, leading to few updates after many rounds. For the comparison of our algorithm, we maintain the learning rate, policy, and exploration size identical to those of SS-SARSA.

Another algorithm is the $d$RGP-TS algorithm (Pike-Burke & Grunewalder, 2019). This approach utilizes GP regression for each arm to estimate the reward distribution. After sampling rewards from the distribution of each arm for predetermined $d$ rounds, the agent pulls the sequence of arms with the highest total rewards. Due to the $K^d$ combinations to draw arms for each $d$ round, a large $d$ becomes unrealistic [4]. Therefore, in our experiments, we consider $d = 1, 2$. This approach helps to avoid the state-combination problem in Q-learning/SARSA and shows a Bayesian regret upper bound. However, the upper bound is derived by repeatedly applying the $d$-step regret. In this simulation, We evaluate full-horizon regret that is defined in metrics. Additionally, we use an alternative implementation that is equivalent to the original but reduces computational complexity. The pseudo-code details and parameter settings are provided in Appendix A.

## 6.2 Simulation results

We show only the results with discounted rewards here, and the undiscounted cases will be deferred to Appendix B.

### 6.2.1 Small-scale problem

For each reward distribution, Figure 1 shows the cumulative regret transitions (left), box plots of cumulative regret in the final round (middle), and the rate of the optimal policy (right).

In the cumulative regret (left), the solid line and the filled area represent the median and the 90% confidence interval, calculated with a thousand simulations. Exploration of each algorithm increases the cumulative regret in the early stage, and then the agent exploits the learned policy. In the box plots of cumulative regret, the black circle, the boxes, the bottom lines in the boxes, the middle lines in the boxes, and the top

---

[3]Even if we set the initial state as $s_{\max} - 1$, after the state transition, each arm state is either 1 or $s_{\max}$. Thus, in the second round, the same problem occurs in the case of $\mathbf{s}_0 = \mathbf{s}_{\max}$.

[4]Pike-Burke & Grunewalder (2019) introduces a computationally efficient algorithm for searching an optimal sequence of arms under large values of $K$ and $d$. However, experimental results showed similar performance for $d = 1, 3$. Additionally, in the regret comparison with other algorithms, only the case $d = 1$ is considered.

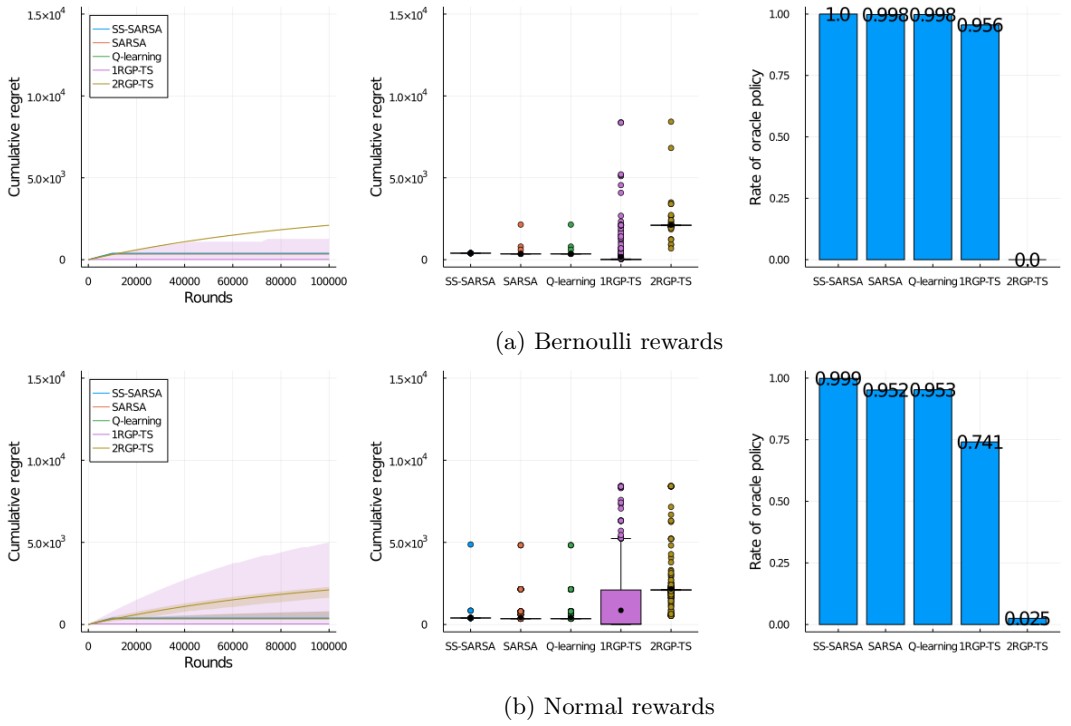

Figure 1: Small-scale problem ($K = 3, s_{max} = 3, \gamma = 0.99999$)

lines in the boxes represent the mean, 50% box, the 25th, 50th, and 75th percentiles, respectively. All points outside the interquartile range of the box, which is 1.5 times the difference between the top and bottom lines, correspond to outliers. The upper and lower lines represent the maximum and minimum values after removing outliers. These graphs reflect the stability of the algorithms through the variation in cumulative regret. The rate of optimal policy over a thousand simulations assesses the quality of exploration.

Figure 1 shows the results of $K = 3$ and $s_{\max} = 3$. In the case of Bernoulli rewards (1(a)), all the methods except 2RGP-TS generally achieve the optimal policy due to the simple setting of the problem. 1RGP-TS shows slightly unstable results. 2RGP-TS does not perform well, suggesting that this method fails to identify the optimal sequence of selected arms even when considering a limited number of combinations.

### 6.2.2 Monotone-increasing rewards

In Figures 2 and 3, the type and arrangement of the graphs are the same as before.

Figure 2 shows the results of $K = 6$. With a larger state space, the standard Q-learning and SARSA do not achieve the optimal policy. For both 6-Hetero and 6-Homo, SS-SARSA stably achieves the optimal policy in most trials within the exploration phase, as seen by no increase of regret after that round. In contrast, in 6-Hetero, 1RGP-TS, the rate of optimal policy is lower. This is caused by the occasional failure to find the optimal policy, which can be seen by the increase of cumulative regret, especially notable in normal rewards. Similarly, in 6-Hetero, 2RGP-TS tends to fail in finding the optimal policy for the same reason as the experiments with $K = 3$.

By comparing the results with $K = 3$ and $K = 6$, we can see that SS-SARSA as well as 1RGP-TS can handle the large state space much more efficiently than $Q$-learning and SARSA, which suffer from complexity even with $K = 6$.

Figure 3 depicts the results for 10 arms. The results of SARSA and Q-learning are not included due to their requirement of a large memory for Q-functions and poor performance in the case of 6 arms. In 10-Hetero, for both Bernoulli (a) and normal (b) rewards, SS-SARSA has low regret and is competitive with 1RGP-TS in

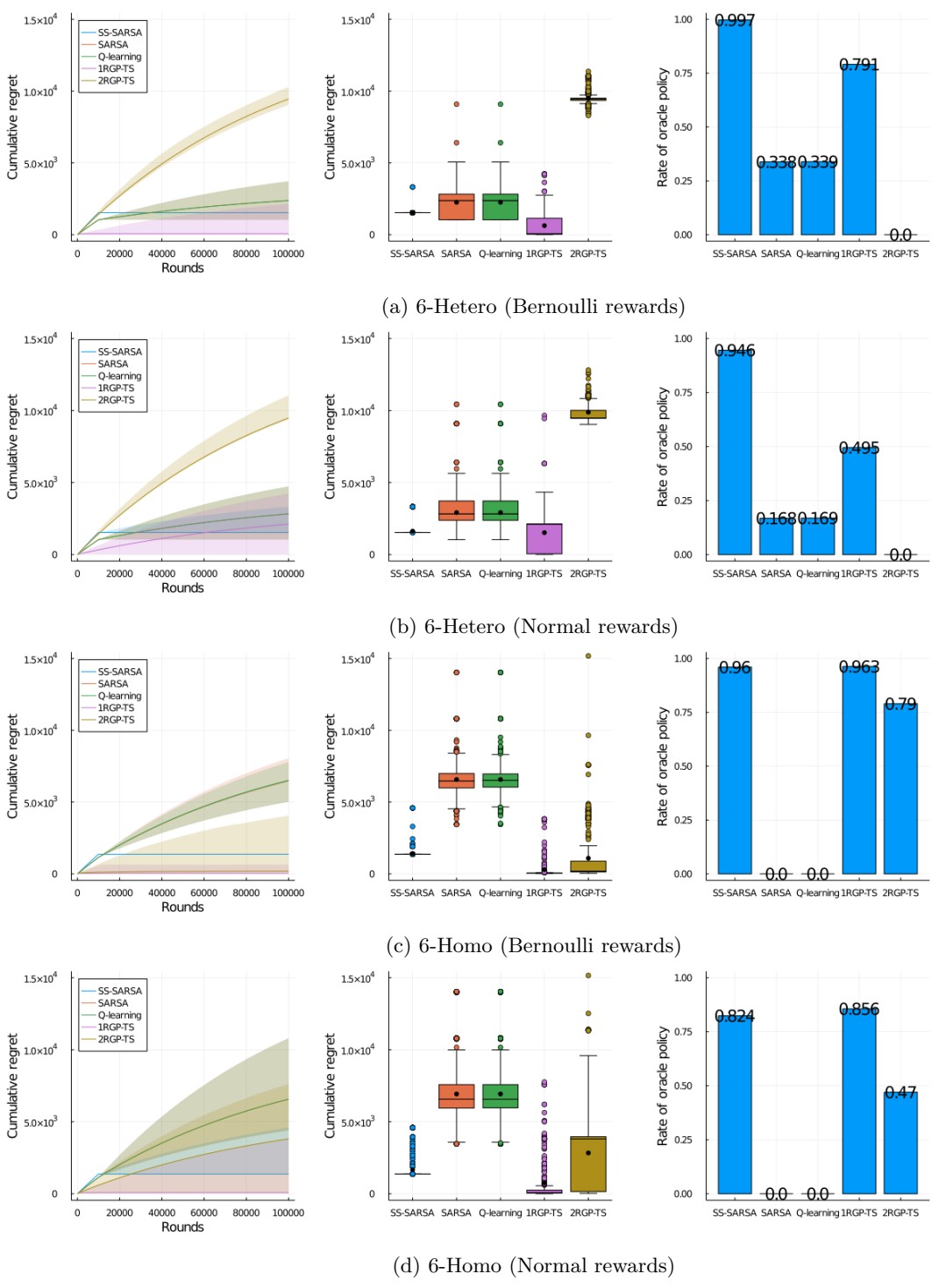

(a) 6-Hetero (Bernoulli rewards)

(b) 6-Hetero (Normal rewards)

(c) 6-Homo (Bernoulli rewards)

(d) 6-Homo (Normal rewards)

Figure 2: Increasing rewards ($K = 6, \gamma = 0.99999$)

obtaining the optimal policy for all simulations. However, While SS-SARSA has a higher rate of the optimal policy and more stable cumulative regret than 1RGP-TS. 2RGP-TS performs the worst in a manner similar to the previous cases. In 10-homo, our algorithm has larger regret than 1RGP-TS due to the exploration phase, but the rate of optimal policy remains competitive.

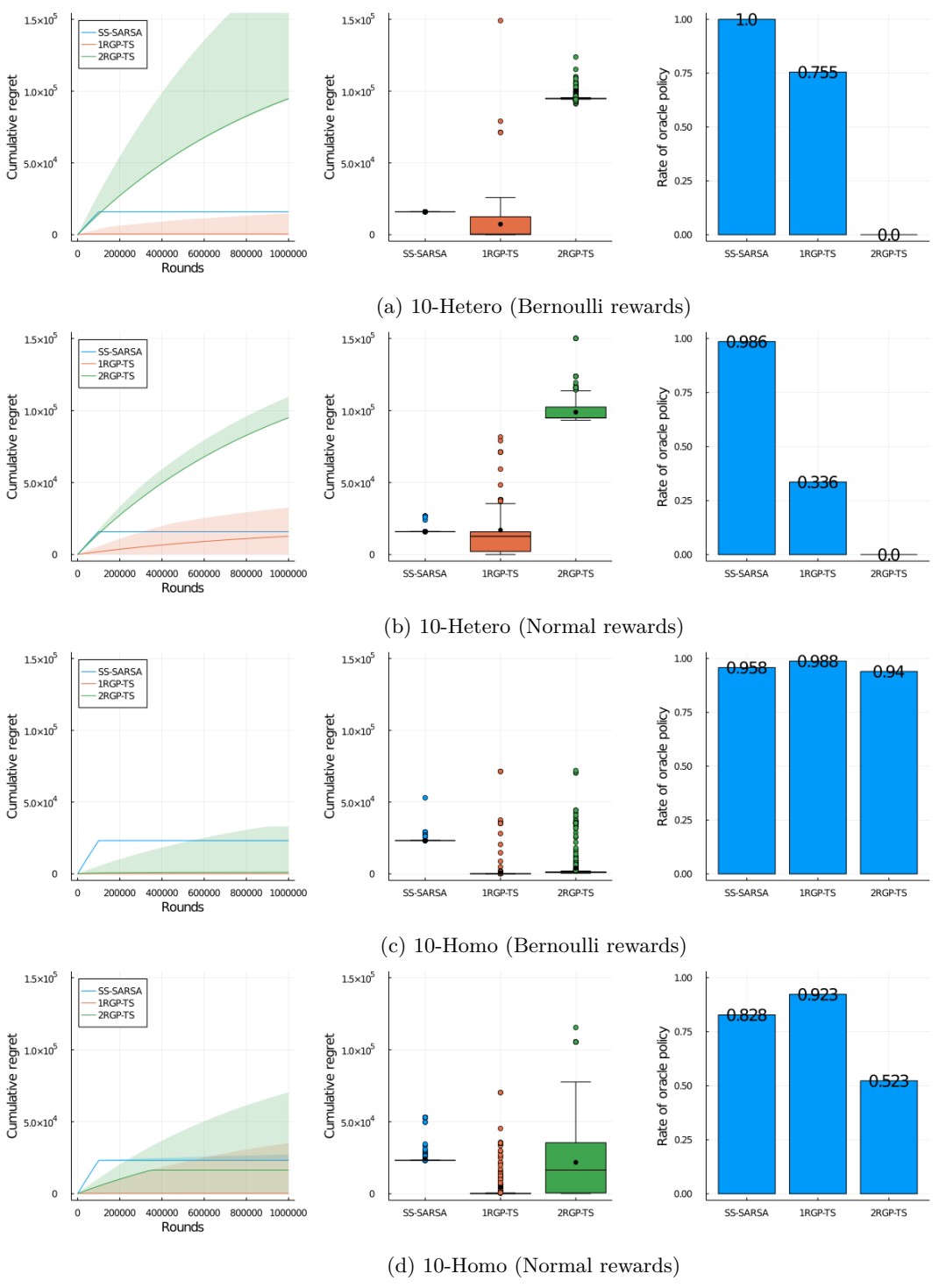

Figure 3: Increasing rewards ($K = 10, \gamma = 0.999999$)

The obtained results indicate that SS-SARSA is the most stability for any case. Notably, in heterogeneous rewards, our algorithm demonstrates the most stable performance with low cumulative regret and the highest rate of optimal policy. 1RGP-TS performs slightly better than our algorithm in the case of homogeneous rewards, but it becomes unstable when dealing with heterogeneous and high-variance rewards.

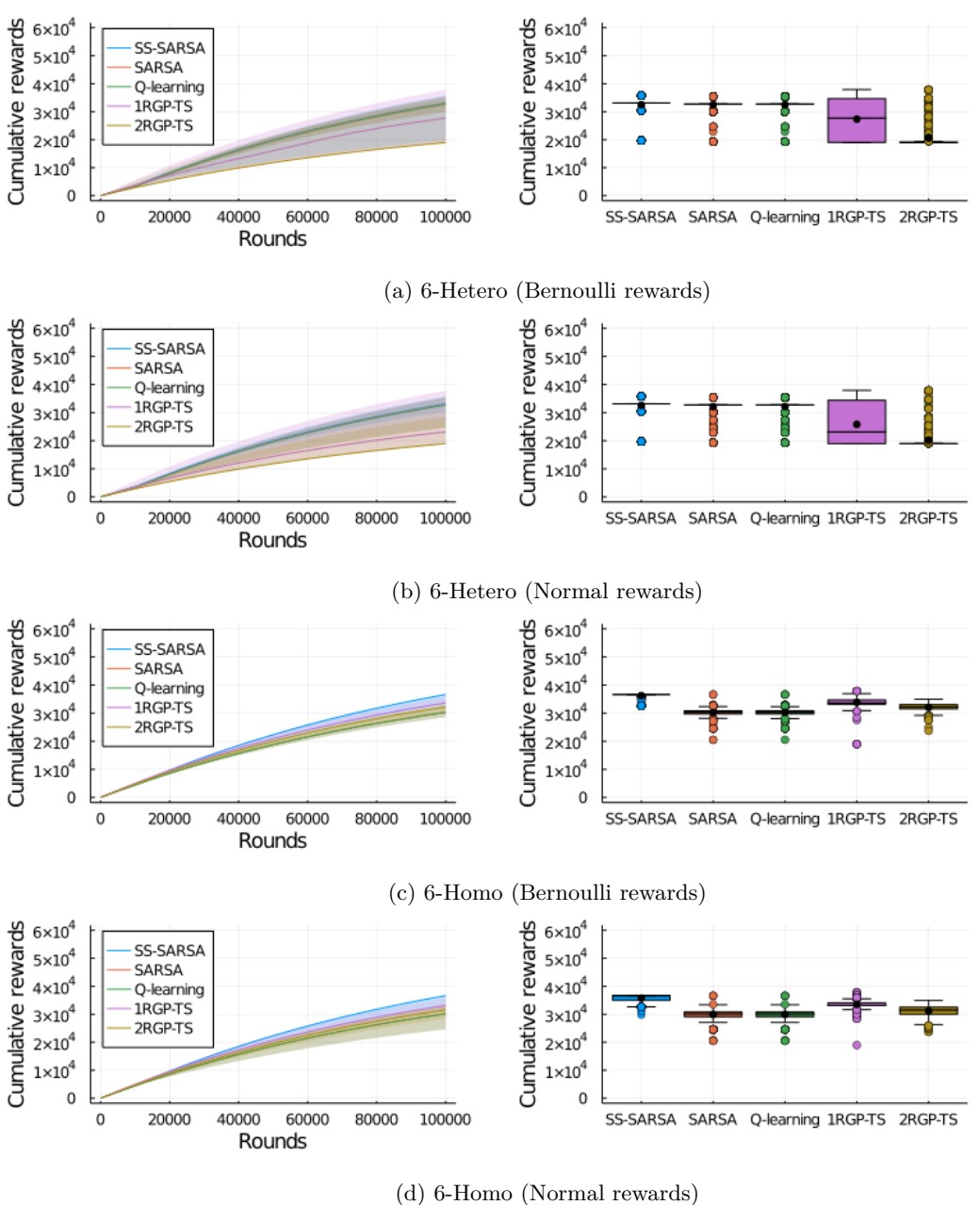

(a) 6-Hetero (Bernoulli rewards)

(b) 6-Hetero (Normal rewards)

(c) 6-Homo (Bernoulli rewards)

(d) 6-Homo (Normal rewards)

Figure 4: Increasing-then-decresing rewards ($K = 6, \gamma = 0.99999$)

### 6.2.3 Increasing-then-decreasing rewards

First, note that in the case of increasing-then-decreasing rewards, we employ the cumulative rewards instead of regret in the left and center graphs of Figures 4 and 5, so the larger values are better.

Those results show that the proposed SS-SARSA performs better or competitively compared to the other methods. In contrast to the monotone rewards, SS-SARSA does not have higher cumulative regret than 1RGP-TS even in the case of homogeneous rewards. The reason for this is the proposed policy; Uniform-Explore-First enforces each SS-Q-function to update uniformly across all states, while the exploration in 1RGP-TS does not take into account the structure inherent in our MDP.

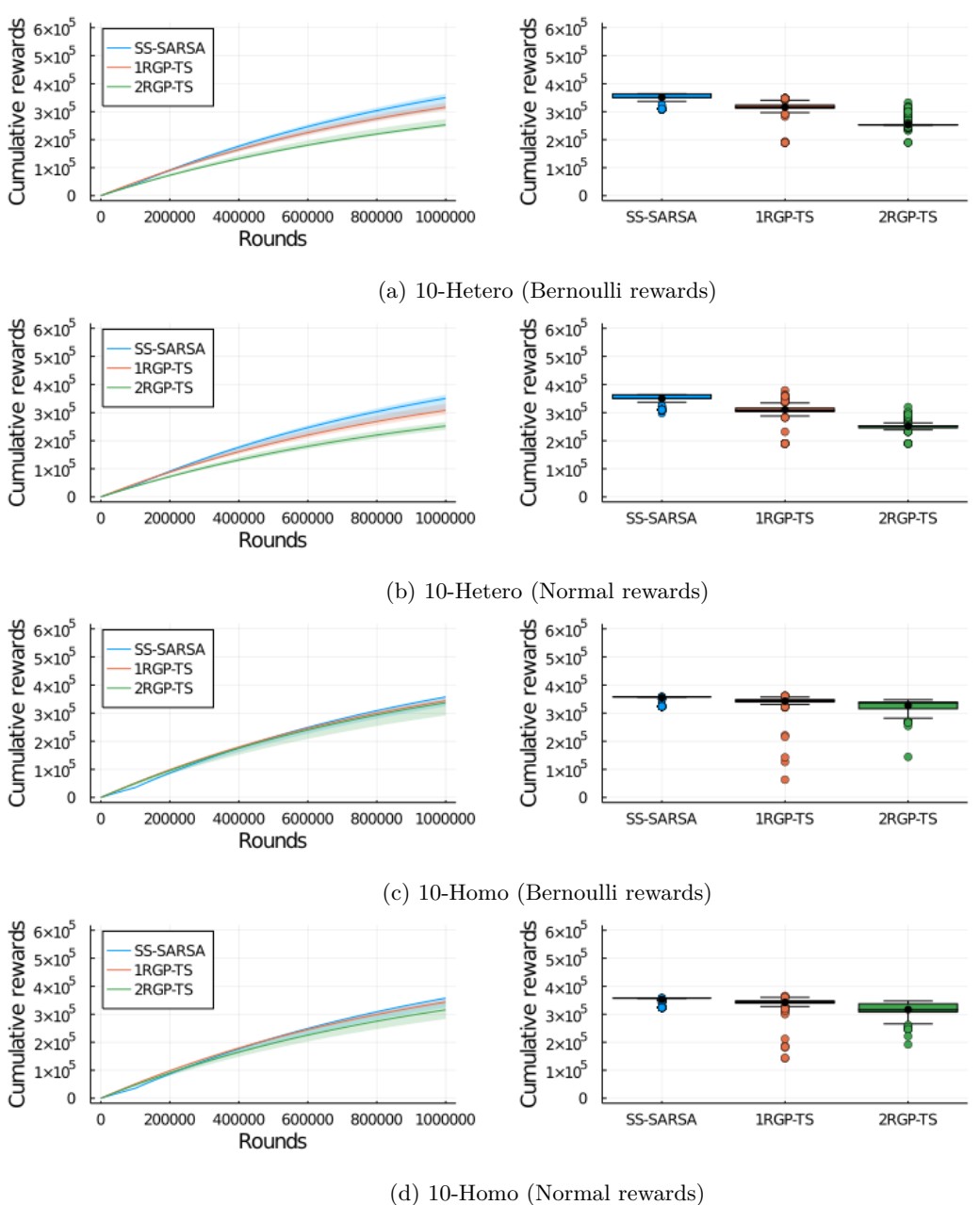

(a) 10-Hetero (Bernoulli rewards)

(b) 10-Hetero (Normal rewards)

(c) 10-Homo (Bernoulli rewards)

(d) 10-Homo (Normal rewards)

Figure 5: Increasing-then-decresing rewards ($K = 10, \gamma = 0.999999$)

Overall, SS-SARSA outperforms other algorithms in terms of stability of regret and rate of optimal policy, regardless of reward distribution, heterogeneity, and state cardinality. The only algorithm that comes close to SS-SARSA is 1RGP-TS, which is competitive or slightly superior in cases of homogeneous and monotone-increasing rewards. However, its performance significantly decreases with heterogeneous or non-monotone rewards.

# 7   Conclusion

We propose an RL algorithm, called SS-SARSA, to solve the recovering bandit problem. This algorithm estimates Q-functions by combining SS-Q-functions and updates like SARSA, leading to efficient learning

and low time complexity. We prove the convergence theorem for the optimal policy. Furthermore, our algorithm performs well in both monotone and non-monotone reward scenarios, as demonstrated through simulations.

The algorithm has several advantages, but it also has some limitations. Firstly, when there are many arms and a large $s_{\max}$, even our algorithm struggles with too many combinations. In such cases, a functional approximation of Q-function is considered for efficient learning. Secondly, we only presented results from simulations, not from real-world data. Our settings require a substantial amount of data points for a person, but to our knowledge such data do not exist. The final is a finite sample analysis. Regret bounds and sample complexity are needed without relying on strong reward structures. These points are left in future works.

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

# A    Alternative Implementation of $d$RGP-TS

This section details the alternative implementation of $d$RGP-TS and parameter settings.

Before introducing the alternative, we briefly review the GP update in $d$RGP-TS algorithm (Pike-Burke & Grunewalder, 2019) [5]. The original paper formulates the reward mechanism as $r(s_a, a) = f_a(s_a) + \epsilon$ where $f_a$ is an unknown function depending on the state for arm $a$ and $\epsilon \sim \mathcal{N}(0, \sigma^2)$ is a Gaussian noise ($\sigma$ is known). When we set a GP prior on $f_a$ and receive $N$ observed rewards $\mathbf{R}_{a,N} = (r_1, r_2, \cdots, r_N)^T$ and states $\mathbf{S}_{a,N} = (s_1, s_2, \cdots, s_N)$ for arm $a$, its posterior is also GP. That is, for $\mathbf{k}_{a,N}(s) = (k(s_1, s), k(s_2, s), \cdots, k(s_N, s))^T$ and positive semi-definite kernel matrix $\mathbf{K}_{a,N} = [k(s_i, s_j)]_{i,j=1}^N$, the mean and covariance of the posterior after $N$ observations are,

$$\mu_a(s; N) = \mathbf{k}_{a,N}(s)^T (\mathbf{K}_{a,N} + \sigma^2 I)^{-1} \mathbf{R}_{a,N} \quad k_a(s, s'; N) = k(s, s') - \mathbf{k}_{a,N}(s)^T (\mathbf{K}_{a,N} + \sigma^2 I)^{-1} \mathbf{k}_{a,N}(s'). \tag{13}$$

Moreover, when $s = s'$, $k_a(s, s'; N)$ is equivalent to the variance $\sigma_a^2(s; N)$. A bottleneck in the above is the time complexity for computing the inverse matrix. In $d$GRP-TS, for every $d$ rounds, the time complexity for the inverse matrix is $\mathcal{O}(c_t(a)^3)$, where $c_t(a)$ represents the number of times an arm $a$ is pulled up to round $t$. Therefore, the original $d$RGP-TS is impractical for large $T$.

Instead, we introduce an alternative implementation of $d$RGP-TS, which is an equivalent update to (13) but reduces its time complexity. Since the states of each arm only take discrete values from one to $s_{\max}$, we can reformulate the argument of each arm's GP function using a $s_{\max}$ dimensional normal distribution. To see this, we define $f_a$ the reward distribution of reward for arm $a \in [K]$ with $s_{\max}$-dimensional normal distribution as follows.

$$f_a = \begin{bmatrix} f_a(1) \\ f_a(2) \\ \vdots \\ f_a(s_{\max}) \end{bmatrix} \sim \mathcal{N} \left( \underbrace{\begin{bmatrix} \mu_{a,1} \\ \mu_{a,2} \\ \vdots \\ \mu_{a,s_{\max}} \end{bmatrix}}_{:=\mu_{a,s_{\max}}}, \underbrace{\begin{bmatrix} k_{a,1}^2 & k_{a,12} & \cdots & k_{a,1s_{\max}} \\ k_{a,21} & k_{a,2}^2 & \cdots & k_{a,2s_{\max}} \\ \vdots & \vdots & \ddots & \vdots \\ k_{a,s_{\max}1} & k_{a,s_{\max}2} & \cdots & k_{a,s_{\max}}^2 \end{bmatrix}}_{:=\mathbf{K}_{a,s_{\max}}} \right) \tag{14}$$

, where $f_a(s)$ is reward function for arm $a$ and state $s \in [s_{\max}]$, $\mu_{a,s}$ is the mean for arm $a$ and state $s$ and $k_{a,ss'}$ is the covariance of the state $s$ and $s'$ for arm $a$ (it is also variance when $s = s'$). Additionally, we define s-th order column of $K_{a,s_{\max}}$ as $k_{a,s_{\max}}(s)$.

Next, we will explain how to update the posterior to reduce the time complexity. When using discrete input, we compute only the $s_{\max}$-dimensional inverse matrix to update the posterior as in (13). To do so, we introduce the count matrix for arm $a$, $C_a$, which is $N \times s_{\max}$ matrix, and its $(i, j)$ element is one when the agent pulls the arm $a$ at state $j \in [s_{\max}]$ for the $i$-th time, otherwise zero. Then, we can easily verify that $\mathbf{k}_{a,N}(s)^T = \mathbf{k}_{a,s_{\max}}(s)^T C_a^T$ and $\mathbf{K}_{a,N} = C_a \mathbf{K}_{a,s_{\max}} C_a^T$. Thus, if the reward were deterministic, $\mu_a(s; N)$ in

---

[5]Some notations differ from the original (Pike-Burke & Grunewalder, 2019) to match the notations of our paper.

(13) could be rewritten as follows.

$$
\begin{aligned}
\mu_a(s; N) &= \mathbf{k}_{a,N}(s)^T (\mathbf{K}_{a,N} + \sigma^2 I)^{-1} \mathbf{R}_{a,N} \\
&= k_{a,s_{\max}}^T C_a^T (C_a \mathbf{K}_{a,s_{\max}} C_a^T + \sigma^2 I)^{-1} \mathbf{R}_{a,N} \\
&= k_{a,s_{\max}}^T C_a^T \underbrace{(C_a \mathbf{K}_{a,s_{\max}} C_a^T + \sigma^2 I)^{-1}}_{\textcircled{1}} C_a \overline{\mathbf{R}}_{a,s_{\max}},
\end{aligned}
\tag{15}
$$

where $\overline{\mathbf{R}}_{a,s_{\max}}$ is the mean reward vector from one to $s_{\max}$ for arm $a$ and we use $\mathbf{R}_{a,N} = C_a \overline{\mathbf{R}}_{a,s_{\max}}$. In practice, since the reward is stochastic, the equality in (15) is replaced by approximation, yet such an approximation saves memory for $R_N$. After applying Sherman–Morrison–Woodbury formula in $\textcircled{1}$,

$$
\textcircled{1} = \frac{1}{\sigma^2} I - \frac{1}{\sigma^2} C_a (\mathbf{K}_{a,s_{\max}}^{-1} + \frac{1}{\sigma^2} C_a^T C_a)^{-1} \frac{1}{\sigma^2} C_a^T.
\tag{16}
$$

Thus, by (16), we can rewrite (15) as follows.

$$
\begin{aligned}
\mu_a(s; N) &= k_{a,s_{\max}}^T C_a^T \left\{ \frac{1}{\sigma^2} I - \frac{1}{\sigma^2} C_a (\mathbf{K}_{a,s_{\max}}^{-1} + \frac{1}{\sigma^2} C_a^T C_a)^{-1} \frac{1}{\sigma^2} C_a^T \right\} C_a \overline{\mathbf{R}}_{a,s_{\max}} \\
&= k_{a,s_{\max}}^T \left\{ \frac{1}{\sigma^2} C_a^T C_a - \frac{1}{\sigma^2} C_a^T C_a (\mathbf{K}_{s_{\max}}^{-1} + \frac{1}{\sigma^2} C_a^T C_a)^{-1} \frac{1}{\sigma^2} C_a^T C_a \right\} \overline{\mathbf{R}}_{a,s_{\max}} \\
&= k_{a,s_{\max}}^T \left\{ \mathbf{C}_{a,\sigma} - \mathbf{C}_{a,\sigma} (\mathbf{K}_{a,s_{\max}}^{-1} + \mathbf{C}_{a,\sigma})^{-1} \mathbf{C}_{a,\sigma} \right\} \overline{\mathbf{R}}_{a,s_{\max}},
\end{aligned}
\tag{17}
$$

where $\mathbf{C}_{a,\sigma} := \frac{1}{\sigma^2} C_a^T C_a$. Therefore, we need to compute only $s_{\max}$-dimensional inverse matrix for updating the posterior mean. In the same way, the posterior covariance matrix can be updated as follows.

$$
k_a(s, s'; N) = k(s, s') - k_{a,s_{\max}}(s)^T \left\{ \mathbf{C}_{a,\sigma} - \mathbf{C}_{a,\sigma} (\mathbf{K}_{a,s_{\max}}^{-1} + \mathbf{C}_{a,\sigma})^{-1} \mathbf{C}_{a,\sigma} \right\} k_{a,s_{\max}}(s').
\tag{18}
$$

The pseudo-code for the algorithm is provided in Algorithm 2. The inputs are total rounds $T$, a standard error of the noise in GP regression $\sigma > 0$, a length scale of RBF kernel $c > 0$, and a size of lookahead $d$. With initial prior with a mean set to zero and covariance matrix with RBF kernel, and initial states, the algorithm proceeds as follows. For every $d$ round, the agent selects combinations of arms $I_{d,t}$ to maximize the total reward, $\sum_{i=0}^{d-1} r(s_{a^{(i)}}^{(i)}, a^{(i)})$, where $s_{a^{(i)}}^{(i)}$ and $a^{(i)}$ represent the state for arm $a^{(i)}$ and arm after $i$ steps of $s$ and $a$, respectively (for $i = 0$, $s_{a^{(0)}}^{(0)} = s_a$ and $a^{(0)} = a$). These values are sampled from normal distributions with estimated means and variances. For the arm $a = I_{d,t}^{(l)}$ selected in $l$th step, the agent receives its reward, updates the posterior mean (17) and covariance (18), and trans to the next state. Since we repeat the above procedure for round $t = 1, 2, \cdots, \lfloor T/d \rfloor$, its time complexity is $\mathcal{O}(K^d T)$.

When we run simulations, we set the input parameters as follows. As discussed in Section 6.1, we specify $d = 1, 2$. The remaining parameters, $\sigma = 1.0$ and $c = 2.5$, are consistent with those utilized in the simulation presented in Pike-Burke & Grunewalder (2019).

## B   Simulation Results with Undiscounted rewards

In this section, we show the performance of our algorithm over the competitors in undiscounted cases (i.e. $\gamma = 1$), which is the default setting in MAB. In each case, the rate of the optimal policy is similar to the discounted case, but since the rewards in the later rounds are undiscounted, the cumulative regret/rewards is different. The most notable case is that of increasing-then-decreasing rewards (Figure 9 and 10); from the left and right graphs, the difference in cumulative rewards between SS-SARSA and 1RGP-TS is greater.

---

**Algorithm 2** $d$RGP-TS (Alternative Implementation)

---

**Input:** $T$, $\sigma$: standard error of noise, $c$: length scale of RBF kernel, $d$: size of lookahead

**Initialize:** $\mu_{s_a,a} = 0 \ \forall a \in [K]$, and $\forall s_a \in [s_{\max}]$; $k_a(i,j) = e^{-(i-j)^2/2l^2} \ \forall a \in [K]$ and $\forall i, j \in [s_{\max}]$;

$\mathbf{s} \leftarrow \mathbf{s}_{\max}$

**for** $t = 1, 2, \cdots, \lfloor T/d \rfloor$ **do**

    Pull a $d$-sequence of arms $I_{d,t} = \text{argmax}_{(a,a',\ldots,a'^{(d-1)})} \sum_{i=0}^{d-1} r(s_{a^{(i)}}^{(i)}, a^{(i)})$, where $r(s_{a^{(i)}}^{(i)}, a^{(i)}) \sim$

$N(\mu_{s_{a^{(i)}}^{(i)}, a^{(i)}}, k_{a^{(i)}}(s_{a^{(i)}}^{(i)}, s_{a^{(i)}}^{(i)})) \ \forall s_{a^{(i)}}^{(i)} \in [s_{\max}]$ , $\forall a^{(i)} \in [K]$, and $\forall i \in \{0, \ldots, d-1\}$

    **for** $l = 1, 2, \cdots, d$ **do**

        $a \leftarrow I_{d,t}^{(l)}$

        $r \leftarrow r(s_a, a)$

        Update $\mu_{s_a,a}$ using (17)

        **for** $m = 1, 2, \cdots, s_{\max}$ **do**

            Update $k_a(s_a, m)$ using (18)

        **end for**

        $s'_k \leftarrow \begin{cases} 1 & \text{for } k = a \\ \min\{s_k + 1, s_{\max}\} & \text{for } k \in [K] \backslash \{a\} \end{cases}$

    **end for**

**end for**

---

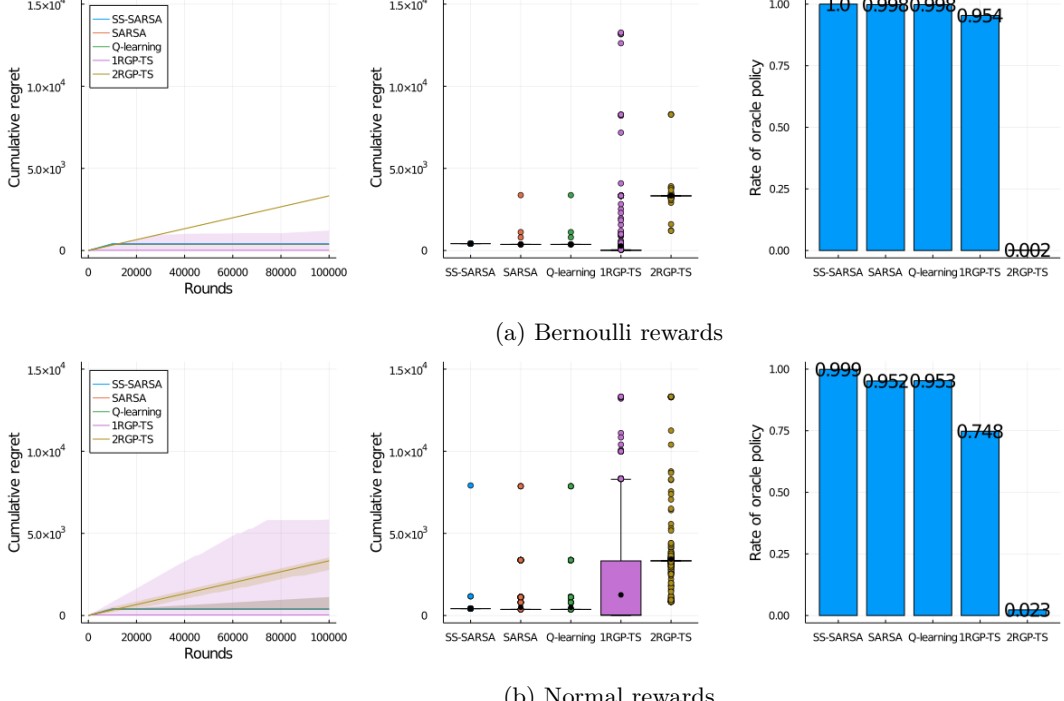

(a) Bernoulli rewards

(b) Normal rewards

Figure 6: Small-scale problem ($K = 3, s_{max} = 3, \gamma = 1$)

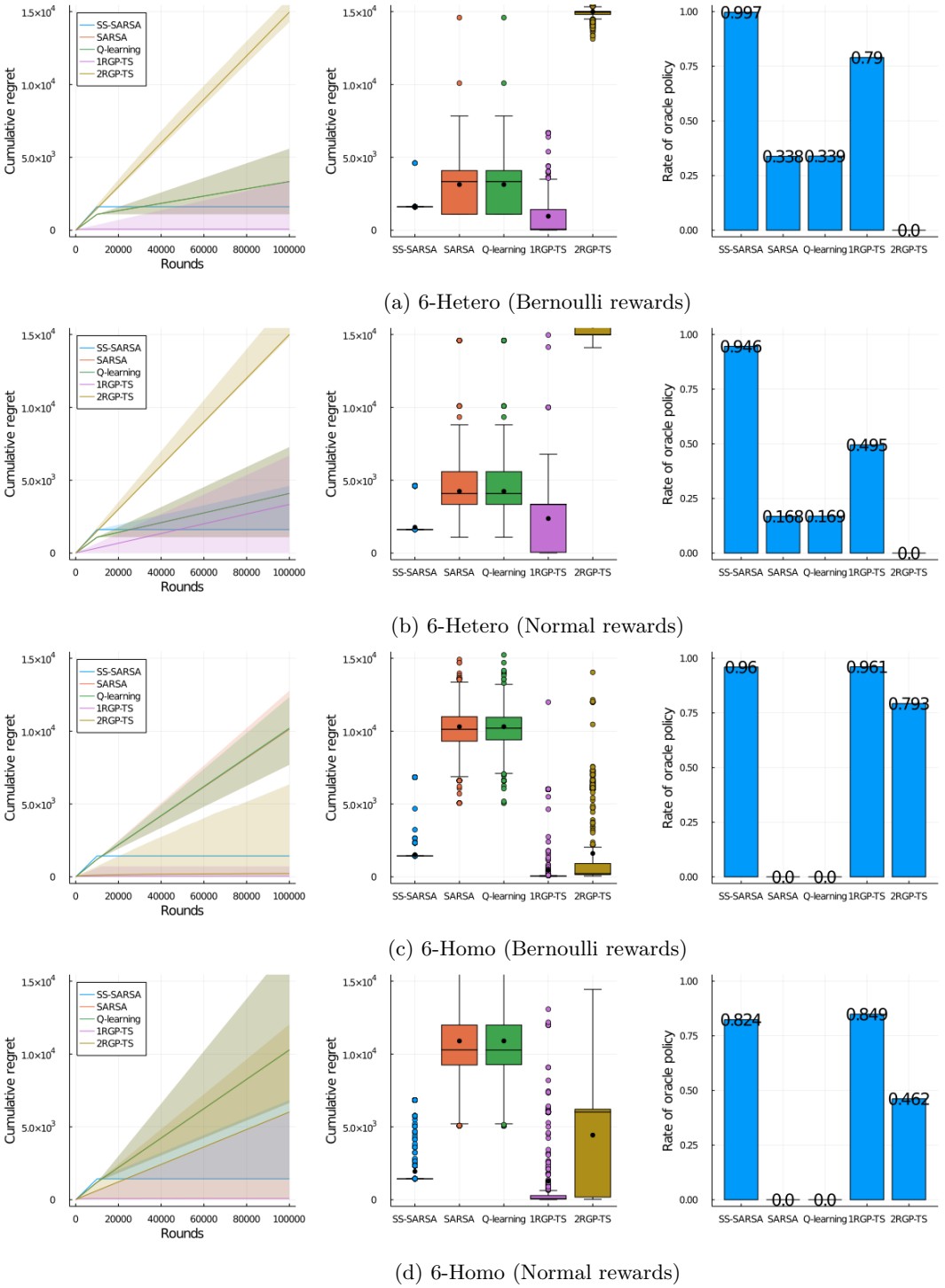

(a) 6-Hetero (Bernoulli rewards)

(b) 6-Hetero (Normal rewards)

(c) 6-Homo (Bernoulli rewards)

(d) 6-Homo (Normal rewards)

Figure 7: Increasing rewards ($K = 6, \gamma = 1$)

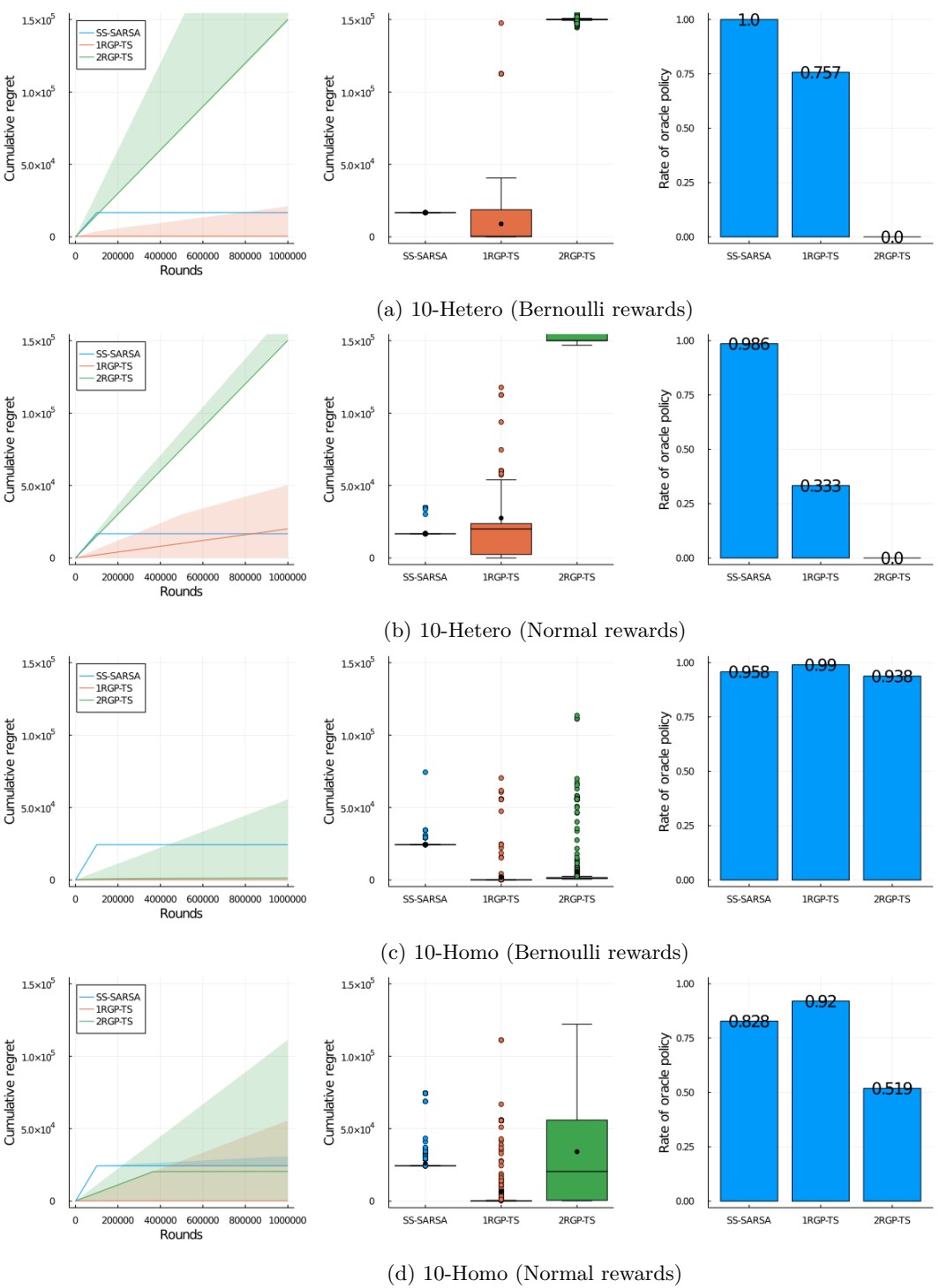

(a) 10-Hetero (Bernoulli rewards)

(b) 10-Hetero (Normal rewards)

(c) 10-Homo (Bernoulli rewards)

(d) 10-Homo (Normal rewards)

Figure 8: Increasing rewards ($K = 10, \gamma = 1$)

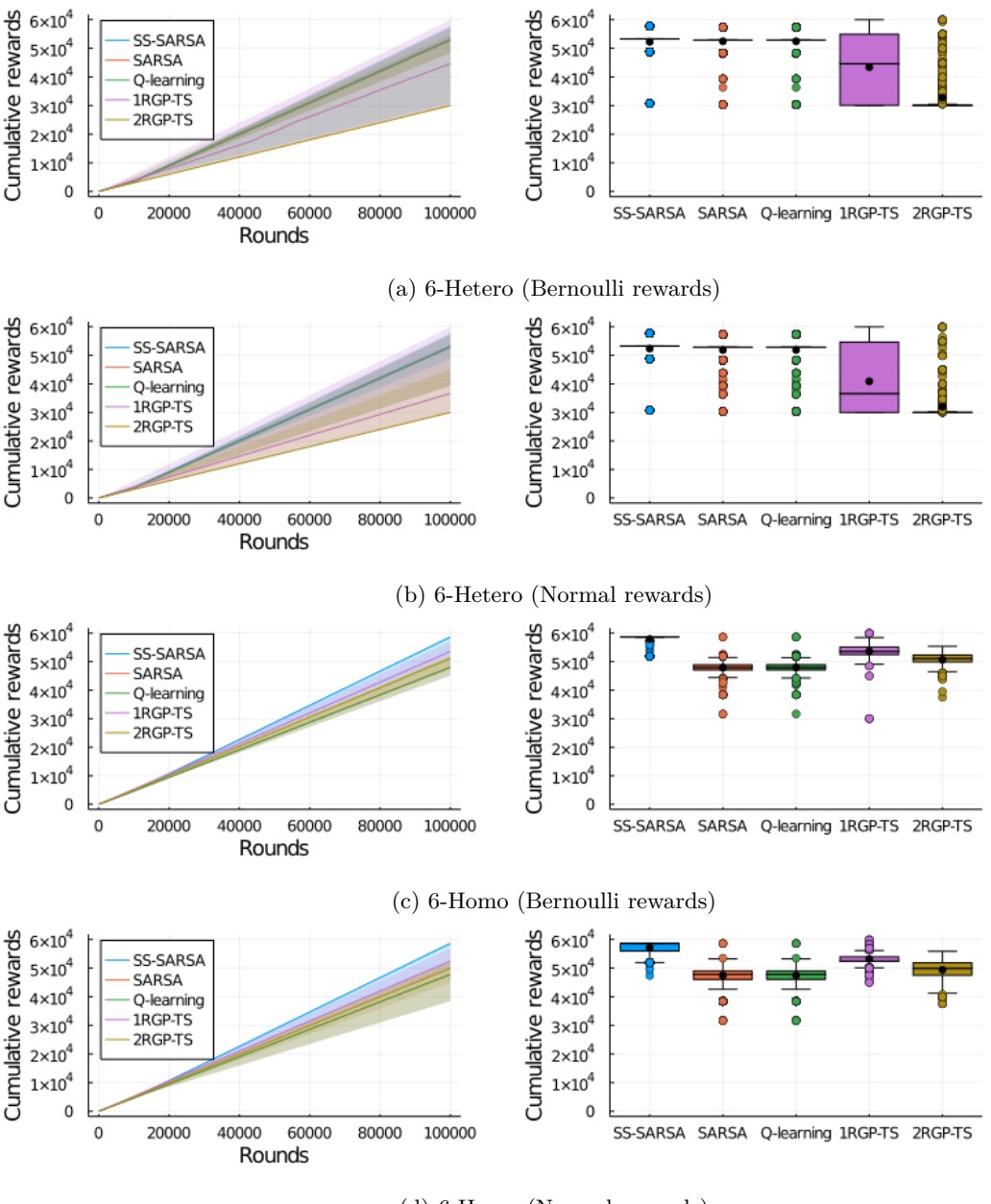

(a) 6-Hetero (Bernoulli rewards)

(b) 6-Hetero (Normal rewards)

(c) 6-Homo (Bernoulli rewards)

(d) 6-Homo (Normal rewards)

Figure 9: Increasing-then-decresing rewards $(K = 6, \gamma = 1)$

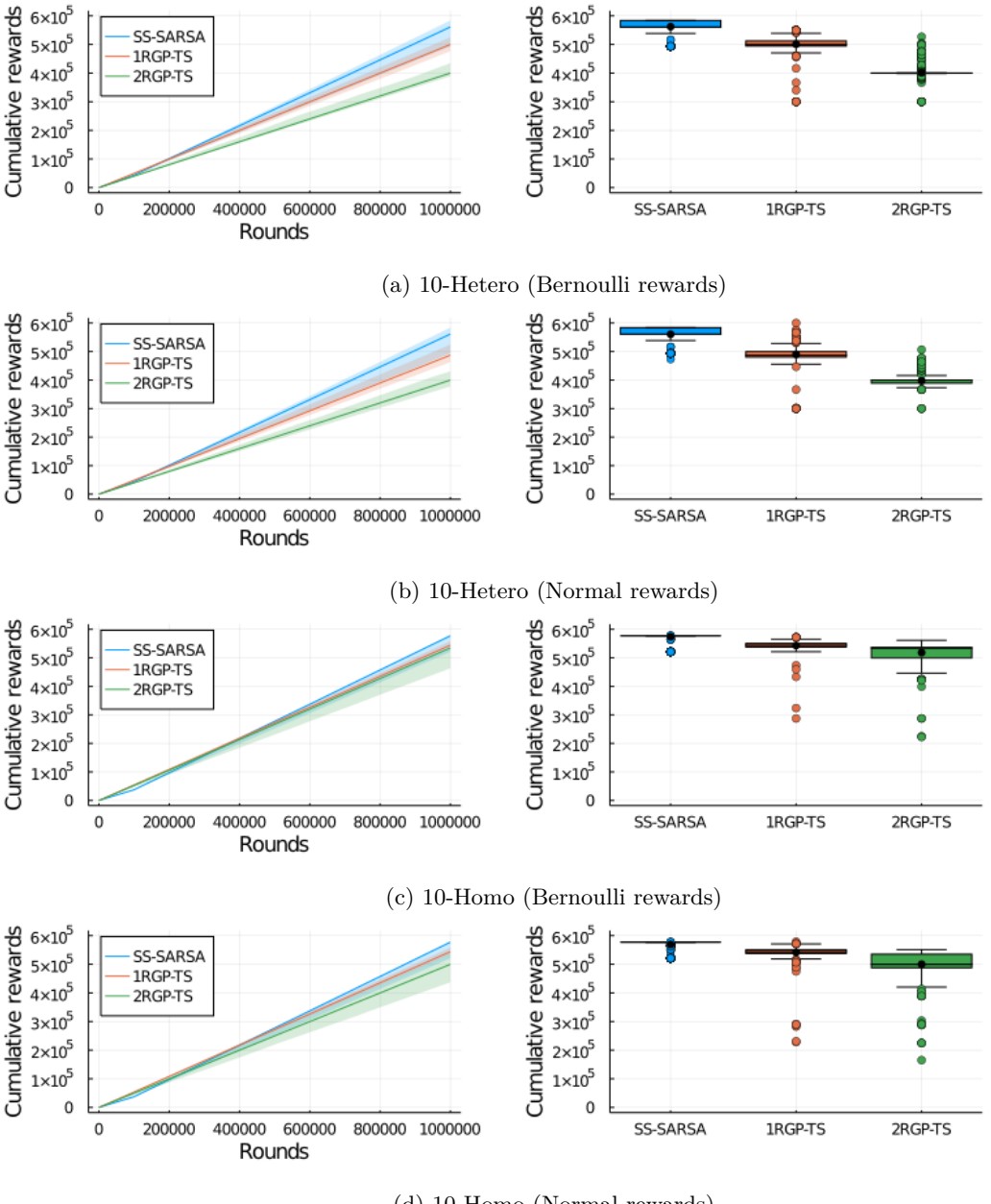

(a) 10-Hetero (Bernoulli rewards)

(b) 10-Hetero (Normal rewards)

(c) 10-Homo (Bernoulli rewards)

(d) 10-Homo (Normal rewards)

Figure 10: Increasing-then-decresing rewards ($K = 10, \gamma = 1$)

