# OpenReview forum: "State-Separated SARSA: A Practical Sequential Decision-Making Algorithm with Recovering Rewards"
_TMLR — Rejected by TMLR_

### Review · Reviewer_zt31 · 2024-04-23

**Summary Of Contributions:**

This paper studies the recovering bandit problem, a variant of multi-armed bandit (MAB) where each arms reward have a "cooldown" since it is last pulled. In order to capture all possible states of the arms, an combinatorially large state space is required to apply a standard approach. Instead this paper proposes to decompose the Q-function into pairs of states so the computational complexity is quadratic rather than exponential in the number of arms.

Experiments were conducted on a set of synthetic setups and compared to some baselines.

**Audience:**

Yes

**Claims And Evidence:**

No

**Requested Changes:**

- Typo: in abstract the proposed approach is named "State-Separate SARSA" instead of "State-Separate**d** SARSA"
- Typo: on page 5 the reference to Section 3 is incorrectly shown as "Chapter 3"
- Page 6: I'm not sure I understand the difference between random exploration and the proposed uniform exploration. In the explanation for random exploration, the paper says "$\pi(a|s)=\frac{1}{K}$ under a random policy" but isn't this just a uniform policy? How is the proposed Uniform-Explore-First different?
- The figures for experimental results are blurry, and many of the lines are blurred together making it hard to tell the difference.
- In some experiments, 1RGP-TS performs just as well as or even better than the proposed SS-SARSA (Fig 2 and 3), but the paper seems to claim their proposed approach is best in all cases.

**Strengths And Weaknesses:**

Strengths
- Simple and intuitive solution to a well described problem.

Weaknesses
- Only simulation experiments are conducted.

---

### Review · Reviewer_MyeZ · 2024-05-02

**Summary Of Contributions:**

This paper study the recovering bandit problem where the reward of each arm depends on the number of rounds elapsed since the last time the arm was pulled. The authors proposed a RL algorithm SS-SARSA to solve this problem, which tries to get around the combinatorial estimation of the Q function. The authors developed asymptotic results for RL algorithms that meet certain conditions; unfortunately, the proposed algorithm doesn't meet all the conditions. The authors simulated experiments and showed their algorithm outperform some baselines.

**Audience:**

Yes

**Broader Impact Concerns:**

N/A.

**Claims And Evidence:**

Yes

**Requested Changes:**

Please address the weaknesses pointed above.

**Strengths And Weaknesses:**

Strengths: This paper is well written and easy to follow. The authors pointed out some failure cases of directly applying RL algorithms on recovering bandits.

Weaknesses:
1. It seems that the proposed algorithm doesn't completely overcome the combinatorial issues: computing and storing the Q function in Eq. (8) requires exponential time and space. The proposed algorithm does need to take argmax wrt the Q function in Eq. (8) when transiting from exploration to exploitation.
2. While the authors developed some theoretical results for RL algorithms that meet certain conditions. The proposed algorithm, unfortunately, doesn't meet all the conditions. As a result, it's not clear if these theoretical guarantees hold true for the proposed algorithm. Additionally, these guarantees are asymptotic in nature; yet finite time guarantees are generally more desired.

---

### Review · Reviewer_mmtP · 2024-05-06

**Summary Of Contributions:**

This paper studies _recovering bandits_, a rested nonstationary setting where an arm's reward depend on the time since last pull. For each arm, this elapsed time between pulls can be modeled via a state-evolution in an MDP, which allows for one to employ techniques from RL. However, the drawback of this model is that there are combinatorially many states (in terms of bandit arms) which calls for more care in designing a practical algorithm.

This paper proposes to fix this issue by working with a newly proposed "state-separated Q-function" which only requires tracking a polynomial number of state-action pairs. Combined with a SARSA-style update rule, this leads to a fairly simple new algorithm for recovering bandits. They show theoretical convergence guarantees of the estimated Q-functions and experiments which show their algorithm is at least on-par with a previous state-of-the-art based on a Gaussian process model, or offers other advantages in performance stability or computational complexity.

**Audience:**

Yes

**Broader Impact Concerns:**

No broader impact concerns.

**Claims And Evidence:**

Yes

**Requested Changes:**

* Can the authors comment on showing regret bounds for their procedure?
* How is the exploration period $E$ set for Theorem 5.1? It seems like it needs to depend on a lot of things about the MDP or Q-functions, in order for the theorem statement to be immediate. There should also be discussion on whether such assumptions that the learner has knowledge of such things is practical or not.
* I don't quite understand the discussion of the two paragraphs following Remark 4.1 on page 6 about why uniform exploration is not suitable for this setting, especially because at the end of the day, the chosen strategy of "Uniform-Explore-First" is still some kind of uniform exploration of arms. I guess one is just saying that the exploration policy should also take into account what the actual state is, so it doesn't allow some state to progress very fast to $s_{\max}$. In particular, I'm not sure what this last sentence "Variation in the frequency of these updates has a negative impact, especially when the state with the highest reward is not $s_{\max}$" is about, and it would be helpful if this could be made into a more precise theorem statement.

**Strengths And Weaknesses:**

Minimizing the combinatorial computation for this model to linear time is an important problem. The state-separated Q-function idea is a neat trick, but I guess not too surprising since one naturally suspects that the complexity of the problem is not truly $s_{\max}^K$ but more like $K \times s_{\max}$ since the states of unplayed arms evolve in a unified way, and this seems to be exactly what is being expressed in keeping track of $(s_k,s_a,a)$ rather than $({\bf s},a)$.

However, I find the main theoretical contribution of this work (Theorem 5.1) underwhelming as it seems to just boil down to a standard convergence result from RL without nearly any modification. There is no discussion of finite-sample or even asymptotic regret bounds, of how the learning rate $\alpha_t$ or exploration period $E$ should be chosen for best regret. Even a rudimentary investigation of regret bounds, making assumptions as needed, would have made the paper stronger in my opinion.

The experiments also do not sell a very convincing message since it seems like the 1RGP-TS algorithm still overall performs as well as (or better in some settings) than SS-SARSA and the main advantages of SS-SARSA is better stability in some settings and computation. If the authors wish to make the paper focus on experiments and application, then I think a more substantive experimental study is in order comparing with other non-stationary algorithms, real-world data, and demonstrations of superiority in computation time.

Thus, I overall feel the paper leaves more to be desired and could use more work (both in discussion and results) before meeting the bar for acceptance.

# Writing Notes
* what is "s+1" in paragraph above display (9)? I presume this is the initial state of the other arms.
* argmin and argmax should be written as \argmin or \argmax in Algorithm 1 pseudocode.
* It might also be good to include some more references on this "rested nonstationary" setting like "Preferences Evolve And So Should Your Bandits: Bandits with Evolving States for Online Platforms" by Khosravi et al., 2024 (and see related works within).

---

### Decision · Action_Editor_7oK9 · 2024-06-04

**Recommendation:** Reject

**Comment:**

This paper proposes an RL algorithm for recovering bandits, where the reward of the arm depends on the time since its last pull. The algorithm is analyzed and empirically evaluated. Both leave a lot of room for improvement. See **Claims And Evidence**. Unfortunately, the authors did not respond with an updated version of the paper. I communicated with them. They confirmed that the paper needs a major revision and that they cannot meet the decision deadline.

**Audience:**

This paper would be of interest to both bandit and RL communities.

**Claims And Evidence:**

This is mixed. The main technical contribution of the paper is an RL algorithm for recovering bandits, where the reward of the arm depends on the time since its last pull. The algorithm is analyzed. The analysis restates a standard result and a more comprehensive analysis, such a regret bound, would strengthen the paper. The algorithm is also empirically evaluated. However, it is not clearly better than the baselines.

**Resubmission Of Major Revision:**

The authors may consider submitting a major revision at a later time.